# SHAPED REWARDS BIAS EMERGENT LANGUAGE

## ABSTRACT

One of the primary characteristics of emergent phenomena is that they are determined by the basic properties of the system whence they emerge as opposed to explicitly designed constraints. Reinforcement learning is often used to elicit such phenomena which specifically arise from the pressure to maximize reward. We distinguish two types of rewards. The first is the base reward which is motivated directly by the task being solved. The second is shaped rewards which are designed specifically to make the task easier to learn by introducing biases in the learning process. The inductive bias which shaped rewards introduce is problematic for emergent language experimentation because it biases the object of study: the emergent language. The fact that shaped rewards are intentionally designed conflicts with the basic premise of emergent phenomena arising from basic principles. In this paper, we use a simple sender-receiver navigation game to demonstrate how shaped rewards can 1) explicitly bias the semantics of the learned language, 2) significantly change the entropy of the learned language, and 3) mask the potential effects of other environmental variables of interest.

## 1 INTRODUCTION

In emergent language research, the goal is to study language as it emerges from the inherent properties of the environment, language, and agents. One pitfall for such experiments, though, is that the language simply mirrors some of design choices of the environment or experimental setting more generally. there is a risk that the language simply mirrors the design choices of the environment. For example, Bullard et al. (2021) introduce a method for discovering optimal languages for communication between independently trained agents, yet rather than emerging from basic principles, the learned language is the result of an intentionally designed search algorithm. Reinforcement learning is a common tool in this field for observing the emergence of language out of a reward maximization pressure. One such design choice which can obscure these emergent properties is adding *shaped rewards* on top of the base reward of the reinforcement learning environment (Wiewiora, 2010).

The base reward of the environment derives directly from succeeding at the task in question. The difficulty with relying solely on the base rewards is that if the task is especially long or complicated, the agent may only receive a reward infrequently which makes for a difficult learning problem. In such a case, base reward is considered *sparse*. This motivates shaped rewards which are inserted at intermediate steps based on domain knowledge in order to introduce an inductive bias towards good solutions. For example, the base reward in chess would simply be winning or losing the game. A shaped reward could then be taking the opponents material while not losing your own. While this shaped reward is often a good heuristic, it can lead to local optima; for example, it discourages strategies which would sacrifice individual pieces in order to win the whole game. While local optima present a problem for maximizing reward, the biases introduced by shaped rewards present a unique problem for emergent language which we will highlight in this paper.

For emergent language research, the inductive bias which shaped rewards introduce is especially problematic because it exerts a significant influence on the learned language whose emergent properties are the object of study. This influence can comprise 1) biasing the semantics of the language, 2) changing a property of the whole language (e.g., language entropy), or 3) masking the influence of some other environmental parameter on the language. For example, some emergent language work incorporates shaped rewards into their environment without accounting for the the biases it may introduce (Mordatch & Abbeel, 2018; Brandizzi et al., 2021; Harding Graesser et al., 2019). From an engineering and design perspective, tweaking the system to achieve a desired result is standard

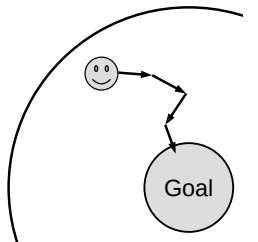 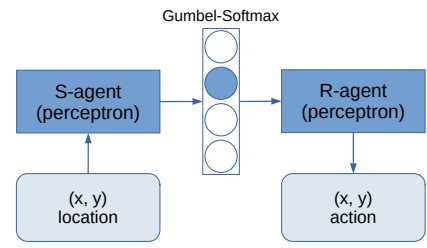

(a) In the centerward environment, the receiver must navigate to the center of the world.

(b) The architecture of our two-agent, asymmetric system.

Figure 1

practice, but from a scientific and experimental perspective, these additional shaped rewards serve as potential confounding factors and hinder accurate observations of the emergent phenomena.

We study this by introducing a simple navigation game with continuous states and actions where a sender passes a one-word message to the receiver (illustrated in Figure 1b and 1a). Within this environment, our experiments look at the entropy and semantics of the emergent language in the absence and presence of shaped rewards. In the course of these experiments, we find that reinforcement learning algorithm's experience buffer size has a significant impact on the entropy of the learned language and potentially explains our experimental findings. To this end we introduce a mathematical model based on the Chinese restaurant process for understanding the effect of experience buffer size on emergent language more generally. We highlight the following contributions in our paper:

- Demonstrating basic ways in which shaped rewards can undesirably influence emergent language experiments
- Presenting a mathematical model for understanding the role of experience buffer size in the entropy of emergent language

## 2 RELATED WORK

### 2.1 TAXONOMY

We intentionally design our environment to study shaped rewards and the entropy of language, which requires it to differ from prior art in specific ways. To elucidate this, we create a taxonomy of emergent language environments based on whether or not the environment has multi-step episodes and the presence of a *trivially optimal* language (defined below). The taxonomy is given in Table 1 and a brief description of each environment is given in Appendix A.

Generally speaking, the motivation for shaped rewards in a given environment is sparsity of the base reward which requires a multi-step, multi-utterance environment. Thus, our experiments naturally require a multi-step environment.

We consider an environment to have a *trivially optimal* language if the information which needs to be communicated from sender to receiver can be perfectly represented in the emergent language. Such a language most frequently arises when the communicated information is derived from a small number of categorical variables encoded in the observation. For example, in an environment where the sender must specify an element of the set $\{red, green, blue\} \times \{square, circle\}$ using messages from the set $\{r, g, b\} \times \{s, c\}$, a trivial language is where color maps to a unique letter in the first position and shape maps to a unique letter in the second position. Other environments have no trivially optimal languages. For example, if the sender must communicate an element of $\{1, 2, ..., 100\}$ using messages from the set $\{a, b, c\}$, there is no trivially optimally language since the sender can at best partition the set of integers to reach an optimal but imperfect solution.

Kharitonov et al. (2020) gives evidence that there is an entropy minimization in pressure inherent in emergent language settings. Building on this, Chaabouni et al. (2021) explicitly look at a the

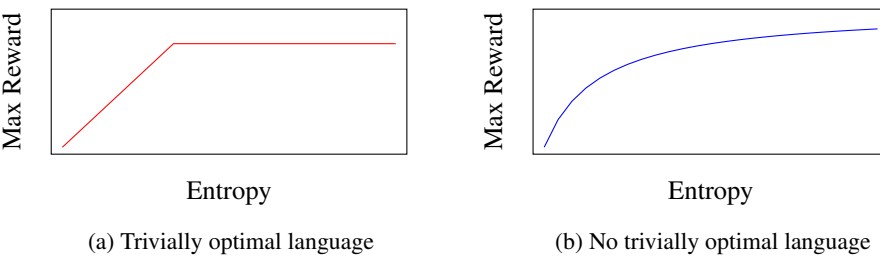

(a) Trivially optimal language      (b) No trivially optimal language

Figure 2: Maximum reward for a given entropy. When a trivially optimal language exists (a), the reward plateaus at a global maximum where further increases in entropy do not increase reward. The environment in this paper has no such trivial language and is more similar (b).

| Paper | Task | Trivial? | Multi-step? |
|---|---|---|---|
| Havrylov & Titov (2017) | image signalling | no | no |
| Kottur et al. (2017) | dialog Q&A | yes | yes |
| Mordatch & Abbeel (2018) | goal specification | yes | yes |
| Lazaridou et al. (2018) | image signalling | yes | no |
| Kharitonov et al. (2020) | vector signalling | yes | no |
| Chaabouni et al. (2021) | color signalling | no | no |
| This paper | navigation | no | yes |

Table 1: Summary of related work in terms of emergent language configurations used.

tradeoff between entropy and reward—higher entropy languages have the potential to convey more information at the expense of being more difficult to learn. This tradeoff disappears if a trivially optimal language is learned since there is no further reward maximization pressure. Language entropy greater than this minimum, then, does not *emerge* from the the reward maximization pressure. Although such an environment does not preclude studying entropy we choose to use an environment where the information being communicated is fully continuous so an increase in entropy can always translate to a increase in reward. This leads to a smooth tradeoff between entropy and reward which is illustrated in Figure 2.

## 2.2 SPECIFIC APPROACHES

The environment and agent configuration of this paper are most closely related to Chaabouni et al. (2021) who test the balance between entropy and semantic precision in a two-agent color discrimination task. Although the color space comprises 330 distinct colors, the environment facilitates languages which cover a (learned) region of nearby colors. In this way, there is no trivially optimal, one-to-one language as the task is inherently fuzzy. Havrylov & Titov (2017) use a signalling game with natural images which lacks a trivially optimal language, but using natural images results in many uncontrolled variables, lessening the ability of the experiments to make basic, first-principles claims.

Studying shaped rewards has been explored previously in emergent language primarily as inductive bias for encouraging some property, such as compositionality, to emerge (Hazra et al., 2021; Jaques et al., 2019; Eccles et al., 2019). This paper, instead, focuses on the negative aspects of inductive biases introduced by shaped rewards regarding how they can hinder empirical investigation of emergent languages.

Superficially, the environment used in this paper bears resemblance to Mordatch & Abbeel (2018) and Kajić et al. (2020) as these both deal with navigation. In both of these environments, the agents communicate about a discrete set of actions or goal locations, whereas the agents in this paper communicate about a continuous action. Neither of the papers specifically investigates the effects of shaped rewards on the languages learned.

## 3   METHODS

### 3.1   ENVIRONMENT

In this paper we use a simple 2-dimensional navigation environment with two closely related tasks. A sender agent observes the position of a receiver agent, sends a message to the receiver, and the receiver takes an action. In the *centerward* task (illustrated in Figure 1a), the receiver is initialized uniformly at random within a circle and must navigate towards a circular goal region at the center. In the *edgeward* task, the receiver is initialized uniformly at random within a circle and must navigate to a goal region comprising the entire area outside of the circle. The centerward environment is a more realistic environment. The edgeward environment, on the other hand, admits a greater variety of languages, since moving consistently in any one direction eventually gets you to the edge; therefore, learning to move in a variety of evenly spaced direction solves the task more quickly but is not strictly necessary. There are no obstacles or walls in the environment. The receiver's location and action are continuous variables stored as floating-point values. If the receiver does not reach the goal region within a certain number of steps, the episode ends with no reward given.

### 3.2   AGENT ARCHITECTURE

Our architecture comprises two agents, conceptually speaking, but in practice, they are a single neural network. The *sender* is a disembodied agent which observes the location of the receiver and passes a message in order to guide it towards the goal. The *receiver* is an agent which receives the message as its only input and takes an action solely based on that message (i.e., it is "blind"). The sender and receiver are randomly initialized at the start of training, trained together, and tested together. The architecture of the agents is illustrated in Figure 1b.

The observation of the sender is a pair of floating-point values representing the receiver's location. The sender itself is a 2-layer perceptron with tanh activations. The output of the second layer is passed to a Gumbel-Softmax bottleneck layer (Maddison et al., 2017; Jang et al., 2017) which enables learning a discrete, one-hot representation, as an information bottleneck (Tishby et al., 2000). The activations of this layer can be thought of as the "words" or "lexicon" of the emergent language. At evaluation time, the bottleneck layer functions deterministically as an argmax layer, emitting one-hot vectors. The receiver is a 1-layer perceptron which receives the output of the Gumbel-Softmax layer as input. The output is a pair of floating-point values which determine the action of the agent. The action is clamped to a maximum step size.

### 3.3   OPTIMIZATION

Optimization is performed using stochastic gradient descent as a part of proximal policy optimization (PPO) (Schulman et al., 2017). Specifically, we use the implementation of PPO provided by Stable Baselines 3 with our neural networks implemented in PyTorch (Raffin et al., 2019; Paszke et al., 2019). Using a Gumbel-Softmax bottleneck layer allows for end-to-end backpropagation, making optimization faster and more consistent than using a backpropagation-free method like RE-INFORCE (Kharitonov et al., 2020; Williams, 1992).

We will give a basic explanation of how PPO (and related algorithms) works as it is necessary to connect it to the mathematical model presented in Section 4. PPO has two stages *sampling* and *optimizing*. During the sampling stage, the algorithm runs the agents in the environment and stores the states, actions, and rewards in a experience buffer (or rollout buffer). The optimization stage then comprises performing gradient descent on the agents (i.e., neural network policy) using the data from experience buffer. The next iteration starts with the updated agents and an empty experience buffer.

### 3.4   REWARDS

We make use of two different rewards in our configuration, a *base* reward and an *shaped* reward. The base reward is simply a positive reward of 1 given if the receiver reaches to the goal region before the episode ends and no reward otherwise. The shaped reward, given at every timestep, is the decrease in distance to the goal. If the goal region is centered at $(0, 0)$, the standard shaped reward for the centerward environment is given by Equation 1; we also use a trivially biased version of the

reward which only takes into account horizontal distance specified in Equation 2.

$$r_t = \sqrt{x_{t-1}^2 + y_{t-1}^2} - \sqrt{x_t^2 + y_t^2} \tag{1}$$

$$r_t' = \sqrt{x_{t-1}^2} - \sqrt{x_t^2} \tag{2}$$

For the edgeward environment, we use the opposite of $r_t$ as the goal is to move *away* from the center.

The interplay between base and shaped rewards is important to understand in the larger context of how reinforcement learning problems are structured. The base rewards are well-motivated and directly correspond to the ultimate aim of the task, but their sparsity can make it difficult for the agents to learn to succeed. Shaped rewards facilitate learning by using expert knowledge to form an inductive bias, yet they present a drawback for traditional reinforcement learning and emergent language. Within reinforcement learning where the goal is to train the best performing agent, shaped rewards can lead to the agent finding local optima if better solutions are excluded by the inductive bias. Within emergent language, the problem is more nuanced as the goal is primarily to study a wide range of *emergent* properties of the language learned within the environment. While base rewards have a natural connection to the environment and task, the shaped rewards introduce a reward signal which is not intrinsically connected with the task and environment, even if it is a good heuristic.

## 4 EXPLANATORY MODEL

We argue that the results presented in Sections 5.2 and 5.3 can be explained by an effective change in the size of PPO's experience buffer. In order illustrate this, we introduce a simple mathematical model based on the Chinese restaurant process (Blei, 2007; Aldous, 1985). While the model does not exactly match our experimental setup, the key similarities allow us to reason about our results as well as potential future experiments.

The Chinese restaurant process is an iterative stochastic process which yields a probability distribution over the positive integers. The key similarity between the Chinese restaurant process and our learning setup is that it they are *self-reinforcing* processes, that is, when a given value is selected in one iteration of the process, the probability that that value is chosen again in subsequent iterations increases. We generalize one aspect of the Chinese restaurant process in Section 4.2 to better match the sampling-optimization procedure of PPO. The primary simplification which this model makes is that it does not take into account the "meaning" of actions and the effects they have within the environment. For example, every successful agent in the centerward environment will use at least three distinct nearly equiprobably so as to span the 2-dimensional space whereas as no such lower bound exists in the stochastic process.

### 4.1 CHINESE RESTAURANT PROCESS

As the name suggests, a useful analogy for the Chinese restaurant process starts with a restaurant with infinitely many empty tables, indexed by the positive integers, which can hold an unbounded number of customers. As each customer walks in, they sit at a populated table with a probability proportional to the number of people already at that table. The customer will sit at a new table proportional to a hyperparameter $\alpha$ which modulates the concentration of the final distribution. The decision the customer makes is equivalent to sampling from a categorical distribution where the unnormalized weights are the customer counts along with the weight, $\alpha$, for the new table. The pseudocode for the Chinese restaurant process is given in Algorithm 1 for $\beta = 1$. By analogy to the neural networks representing our agents, we can view the tables as bottleneck units and the customers choosing a table as parameter updates which reinforce the use of that unit in accordance with the reward. Mordatch & Abbeel (2018) implicitly assume this when they introduce a reward corresponding to the probability that the emergent lexicon is generated by a Chinese restaurant process (a Dirichlet process, in their words).

The self-reinforcing property can be expressed informally as: more popular tables get more new customers, keeping them popular. A higher $\alpha$ means that customers are more likely to sit at a new table, so the distribution over tables will be more spread out in expectation. The distribution stabilizes as the number of iterations goes to infinity as an individual new customers has a diminishing effect the relative size of the weights.

---

**Algorithm 1** Expectation Chinese Restaurant Process

---

```
1   assert type(alpha)   is float and alpha   >  0
2   assert type(n_iters) is int   and n_iters >= 0
3   assert type(beta)    is int   and beta    >  0
4
5   def sample_categorical_alpha(weights):
6     w_alpha = weights.copy()
7     k = num_nonzero(weights)
8     w_alpha[k + 1] = alpha
9     return sample_categorical(w_alpha / sum(w_alpha))
10
11  weights = array([1, 0, 0, ...])
12  for _ in range(n_iters):
13    addend = array([0, ...])
14    for _ in range(beta):
15        i = sample_categorical_alpha(weights)
16        addend[i] += 1 / beta
17    weights += addend
18  return weights / sum(weights)
```

---

## 4.2 EXPECTATION CHINESE RESTAURANT PROCESS

The key difference between how the Chinese restaurant process and PPO works is the relationship between sampling (i.e., simulating episodes) and updating the weights/parameters. In each iteration, the regular Chinese restaurant process draws a sample based on its weights and updates those weight immediately. In PPO, the agent will populate the experience buffer with a number of steps (on the order of 100 to 1000) in the environment before performing gradient descent with that buffer to update the parameters. As a result, the parameter update is performed based on a weighting across multiple bottleneck units based on how often they were used in the episodes recorded in the experience buffer.

Thus, to the appropriately generalize the Chinese restaurant process, we introduce the *expectation Chinese restaurant process*. In this process, we add a hyperparameter $\beta$ which is a positive integer describing how many samples we take from the distribution before updating the weights; the updates are normalized by $\beta$ so the sum of all weights still only increases by 1 per iteration. The restaurant analogy breaks down here as we would have to say that in each iteration, $\beta$ customers simultaneously and independently make a decision, get shrunk to $\frac{1}{\beta}$th their size, and then sit at their table of choice. The pseudocode for the expectation Chinese restaurant process is given in Algorithm 1.

## 5 EXPERIMENTS

Each run of experiment starts by training a sender and receiver for a fixed number of timesteps for a range of independent variable values. The trained models are then evaluated by initializing 3000 episodes at evenly distributed locations using Vogel's method (Vogel, 1979). In most settings, the agents are able to achieve a $100\%$ success rate during training and evaluation; we remove any models which do not from consideration. All model for our experiments use $2^6 = 64$ bottleneck units which translates to a maximum entropy of 6 bits. Hyperparameters are given in Appendix B.

## 5.1 BIASED SEMANTICS

In our first experiment we demonstrate how shaped rewards which are trivially biased directly distort the semantics of the language, that is, the action associated with each bottleneck unit. We compare three environments, no shaped rewards, the standard shaped reward, and the trivally biased shaped reward. We visualize the semantics of the language with so-called "sea star plots" in Figure 3. Each arm of the sea star is the action taken by the receiver in response to a single bottleneck unit with opacity representing the frequency of use.

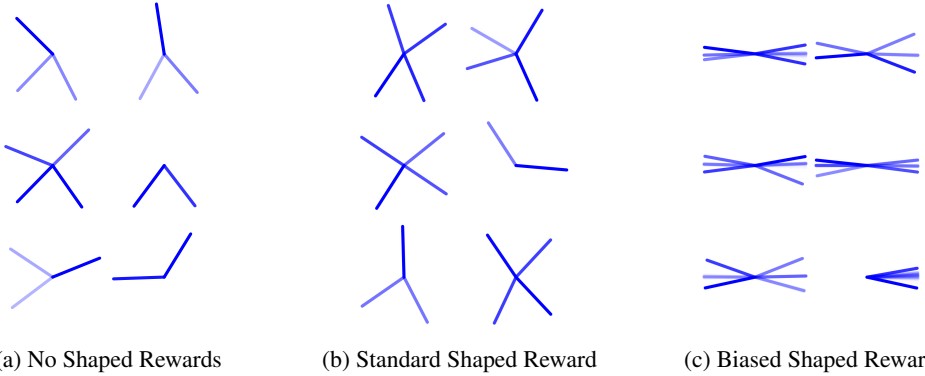

(a) No Shaped Rewards        (b) Standard Shaped Reward        (c) Biased Shaped Reward

Figure 3: Sea star plots for three different settings in the edgeward navigation environment. Each "sea star" corresponds to an independent language learned in the given setting.

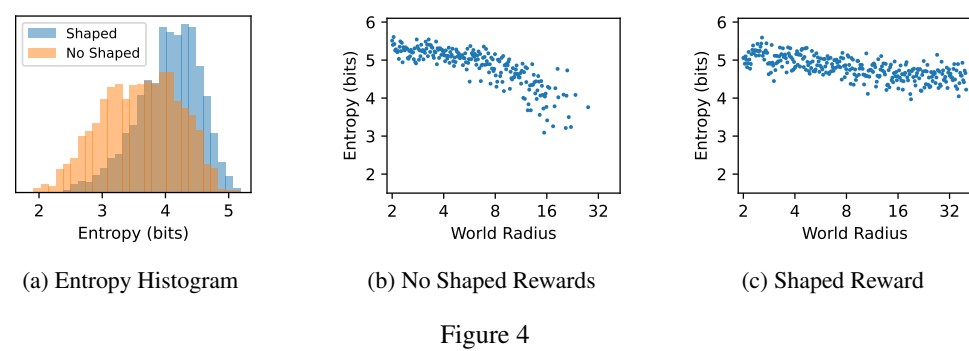

(a) Entropy Histogram        (b) No Shaped Rewards        (c) Shaped Reward

Figure 4

In the setting with no shaped rewards, we see actions (i.e., the meanings of the messages) learned featuring 2 to 4 arms pointing in a variety of directions. Since the standard shaped reward takes both dimensions into account, we do not see any bias in the direction of the learned actions. With the trivially biased reward, though, we see the the learned languages exclusively favor actions near to the horizontal axis. In this setting, nothing explicitly prevents the agents from learning vertical actions, but the fact that horizontal dimensions receive the shaped reward makes those actions easier to learn.

## 5.2 CHANGING THE DISTRIBUTION OF ENTROPY

Naturally, a shaped reward which favors certain actions over others will bias the semantics of the language. Thus, our second experiment investigates more closely the effect that shaped rewards *without* this explicit bias can have. Specifically, we investigate the distribution of language entropies in the two environments. By entropy we are specifically referring to the Shannon entropy (measured in bits) of the bottleneck units as used in the trained agents' language (as averaged over 3000 quasirandomly initialized episodes).

Entropy is an important aspect of language as it represents the upper bound on the information that the language can convey. Languages with higher entropy can yield more precise meaning in their utterances, yet this comes at the cost of being more difficult to learn or acquire as they need a greater variety of training example to be learned.

To investigate the distribution of language entropies, we look at a histogram showing the Shannon entropy of languages belonging to environments with and without shaped rewards. The distributions is computed from 2000 independent runs for each reward setting. This is shown in Figure 4a. The presence of shaped rewards shifts the distribution upwards, demonstrating that even shaped rewards which is free of a trivial bias can still bias the emergent language. A potential explanation of these results is discussed and illustrated in Section 5.4.

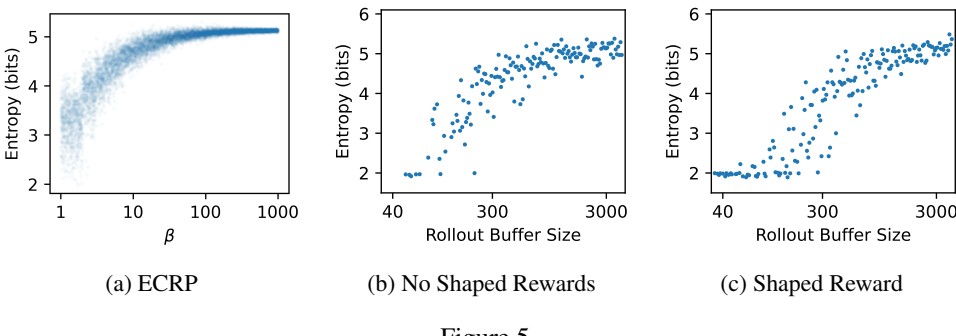

(a) ECRP         (b) No Shaped Rewards         (c) Shaped Reward

Figure 5

## 5.3 MASKING ENVIRONMENTAL PARAMETERS

In our final primary experiment, we demonstrate how shaped rewards can mitigate the influence of environmental parameters on the entropy of the learned language. This is an issue insofar as the presence of shaped rewards make it difficult to observe an emergent property of interest. Specifically, we look at how the standard shaped rewards hide the effect of world radius on entropy in our centerward environment.

In Figures 4b and 4c, we plot the language entropies against different world radii. In both settings, we observe that entropy decreases as the world radius increases, but the setting with no shaped rewards shows a much more rapid decrease in entropy. We offer one possible explanation for this effect in Section 5.4. When the only reward signal agents have access to is the base reward, they can only accomplish the task by finding the goal randomly at first; as the size of the environment increases, the chance of finding the goal with random movements decreases and the agent pair often fails to learn a fully successful language at the highest world radii.

## 5.4 EXPERIENCE BUFFER SIZE

As an explanatory experiment, we demonstrate how changing the size of the PPO experience buffer has a significant impact of the entropy of the emergent language. We compare this with the effects we would expect to see according to the model presented in the next section, i.e., the expectation Chinese Restaurant process. In turn we use this to explain one mechanism by which shaped rewards can have the observed effects on entropy shown by the previous experiments.

In Figure 5a we show the effect of a logarithmic sweep of $\beta$ on entropy of the expectation Chinese restaurant process. We first observe that increasing $\beta$ reduces the variance between distributions yielded by the process since, as $\beta$ increases, the individual updates made in each iterations are also reduced in variance. In fact, in the limiting case as $\beta \to \infty$, the process will always yield the same distribution as the update will just be the expectation of sampling from the categorical distribution described by the (normalized) weights (plus $\alpha$). The second effect is that increasing $\beta$ will decrease the concentration (i.e., increase the entropy), on average, of the distribution yielded from the process. The intuition behind this is that the since each update is less concentrated, the distribution as a whole will be less concentrated as the probability mass will be spread out.

These results can be used to explain, in part, both the effect of shaped rewards and world radius on entropy. First, though, we must establish an analogous correspondence between the expectation Chinese restaurant process and the PPO-learning process. An iteration of the process described in Algorithm 1 consists of sampling from the modified categorical distribution (Line 15) and incrementing the weights (Line 17). In PPO, the sampling corresponds to populating the experience buffer with steps associated with a reward, and the increment operation is analogous to PPO performing gradient descent on the agents using the buffer. Thus, $\beta$ is analogously increased for PPO when the number of successful episodes per iteration which is dependent both on the size of the experience buffer as well as the environmental factors affecting frequency of success.

In Figures 5b and 5c, we directly vary the size of the experience buffer in our environments with and without shaped rewards. Both environments replicate the correlation between $\beta$/buffer size and entropy, though the decrease in variance is less distinct as buffer size increases.

Having established this correlation, we can offer a potential explanation to the experiments involving world radius as well as the distribution of entropies between the environments with and without shaped rewards. Shaped rewards effectively increase $\beta$ since it assigns a reward signal to every step whereas the base reward-only environment requires successful episode. This effect is exacerbated when the world radius is increased, the base reward-only environment yields rewards less frequently in the beginning because randomly finding the goal is less likely. This effectively decreases $\beta$ which corresponds to a lower entropy and higher variance.

## 6 CONCLUSION

We have, then, demonstrated the pitfalls that shaped rewards present for emergent language research: directly biasing the learned semantics of the language, changing the distribution of an emergent property of the language (i.e., entropy), and masking the emergent effects of other environmental variables. These experiments were performed with with a novel navigation-based emergent language environment. This environment allows allows for shaped rewards through multi-step episodes and avoids a trivially optimal language by employing a continuous state and action space. In addition to this, we introduced the expectation Chinese restaurant process both to explain our own experimental results and to provide a foundation for future models of emergent language.

The limitations of this work can be illustrated through the future directions that could be taken. First, studying a variety of environments would further characterize what biases shaped rewards introduce into emergent languages. Second, increasing the complexity of environments would greatly expand the range of both emergent properties and types of shaped rewards which could be studied. For example, a rich environment like chess presents many emergent strategies such as the valuation of positions, balance between offense and defense, and favoring certain pieces; shaped rewards could then take the form of rewarding stochastic strategies, evaluating positions with traditional chess engines, or assigning pieces explicit point values. Furthermore, while we only use the expectation Chinese restaurant process as an explanatory model, further work could design experiments to demonstrate its predictive power.

The studies presented in this paper are both exploratory and anticipatory in nature since emergent language research has yet to tackle environments difficult enough to require shaped rewards. Nevertheless, the field will follow reinforcement learning in tackling progressively more difficult tasks which will present further opportunities for or even require shaped rewards. When this occurs, accounting for the biases which are inherent to shaped rewards is imperative to preserving the integrity of emergent language experiments. This work, then, prepares researchers in the field for these future challenges.

### REPRODUCIBILITY STATEMENT

The code used in association with this paper is located at `https://example.com/repo-name` (in a ZIP file for review). Reproduction instructions are located in the `README.md` file. Experiments were performed on a 20-thread Intel Core i9-9900X server on which they take less than 24 hours to run. No random seeds are recorded or provided as the training process is stable and similar results should be observed for any random seed.

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

## A ENVIRONMENTS OF RELATED WORK

**Havrylov & Titov (2017)** Standard signalling game where the observations are natural images from Microsoft COCO. There is no trivially optimal language since the information being communicated is simply which image is being shown. There are natural image classes (e.g., cat vs. dog), but they are not necessarily the features of the images which the agents need to communicate. The standard signalling game is single-step.

**Kottur et al. (2017)** "Task and talk" dialog game where one agent must ask questions of the other agent to learn the attributes of an object. Specifically, the questioner has a set of attributes it must learn (the objective) about the object that only the answerer can see. The desired language has the questioner using a unique message to specify a property and the answerer responding with a unique message to specify the value for that property such that messages are always in one-to-one correspondence what they are communicating. Such a language is trivially optimal. Multi-round dialog environments are inherently multi-step.

**Mordatch & Abbeel (2018)** Collaborative navigation game where multiple agents tell each other whither to move in a 2D environment. The environment consists of agents and landmarks distinguisted by color. Each agent is given a "goal"; for example, the blue agent might receive "red agent go to green landmark" (represented as a categorical feature vector) which the blue agent then has to communicate to the red agent. The agents have a vocabulary large enough to assign a unique "word" to each concept being expressed in one-to-one correspondence, yielding a trivially optimal language. Each episodes consists of multiple timesteps which can each have their own utterance.

**Lazaridou et al. (2018)** An image-based signalling game using rendered images from MuJoCo. The information being communicated is the shape and color of an object where set of shapes and colors are both small. Since the number of words available is at least as big as the cardinalities of the sets and the sender is able to use multiple words per utterance, it is possible to construct a trivially optimal langauge. The standard signalling game is single-step.

**Kharitonov et al. (2020)** A binary vector-based signalling game with shared context. The goal of the signalling game is to communicate the bits of a binary vector which are *not* shared by the sender and receiver. The messages consist of a single symbol where the number of unique symbols is greater than the combination of bits to communicated, thus there is a trivially optimal langauge where one unique symbol is assigned to each possible combination of unshared bits. The standard signalling game is single-step.

**Chaabouni et al. (2021)** A signalling game with 330 different colors represented as a real vector in CIELAB color space. The game is set up so that colors which are nearby never appear in the same distractor set which encourages solutions which can cover some arbitrary region of colors. Due to this "fuzzy" concept of solution (i.e., not using 330 distinct words for each color), we consider this environment not to have a trivially optimal solution. The standard signalling game is single-step.

## B  HYPERPARAMETERS

### B.1  DEFAULT CONFIGURATION

**Environment**

- Type: centerward
- World radius: 9
- Goal radius: 1
- Max steps per episode: $3 \times$ `world_radius`

**Agent Architecture**

- Bottleneck size: 64
- Architecture; sender is 1-3 and receiver is 5; bottleneck size is $N$
  1. Linear w/ bias: 2 in, 32 out
  2. Tanh activation
  3. Linear w/ bias: 32 in, $N$ out
  4. Gumbel-Softmax: $N$ in, $N$ out
  5. Linear w/ bias: $N$ in, 2 out (action) and 1 out (value)
- Bottleneck (Gumbel-Softmax) temperature: 1.5
- Weight initialization: $\mathcal{U}\left(-\sqrt{\frac{1}{n}}, \sqrt{\frac{1}{n}}\right)$, where $n$ is the input size of the layer (PyTorch 1.10 default)

**Optimization**

- Reinforcement learning algorithm: proximal policy optimization
  - Default hyperparameters used unless otherwise noted: `https://stable-baselines3.readthedocs.io/en/v1.0/modules/ppo.html`
- Training steps: $1 \times 10^5$
- Evaluation episodes: $3 \times 10^3$
- Learning rate: $3 \times 10^{-3}$
- Experience buffer size: 1024

- Batch size: 256
- Temporal discount factor ($\gamma$): 0.9

## B.2 EXPERIMENT-SPECIFIC CONFIGURATIONS

Note that we define a logarithmic sweep from $x$ to $y$ (inclusive) with $n$ steps to be defined by Equation 3.

$$\left\{ x \cdot \left(\frac{y}{x}\right)^{\frac{i}{n-1}} \;\middle|\; i \in \{0, 1, \ldots, n-1\} \right\} \tag{3}$$

**Biased Semantics**

- Type: edgeward
- World radius: 8
- Goal Radius: 8
- Experience buffer size: 256
- Batch size: 256

**Changing the Distribution of Entropy**

- Number of independent runs per configuration: 2000
- Experience buffer size: 256
- Batch size: 256

**Masking Environmental Parameters**

- World radius: logarithmic sweep from 2 to 40 with 300 steps

**Experience Buffer Size**

- Experience Buffer Size: logarithmic sweep from 32 to 4096 with 200 steps (floor function applied as experience buffer size is integer-valued)
- Training steps: $2 \times 10^5$

**Expectation Chinese Restaurant Process**
This is not an emergent language experiment and just consists of running the mathematical model.

- $\alpha$: 5
- $\beta$: logarithmic sweep from 1 to 1000 with 10 000 steps (floor function applied, though plotted without floor function)
- Number of iterations per run: 1000

