# OpenReview forum: "Shaped Rewards Bias Emergent Language"
_ICLR.cc/2022/Conference — ICLR 2022 Submitted_

### Official Review · Reviewer_uFB4 · 2021-10-16

**Correctness:** 3
**Technical Novelty And Significance:** 4
**Empirical Novelty And Significance:** 4
**Recommendation:** 6
**Confidence:** 4

**Main Review:**

## Summary

I really liked this paper. There are a bunch of typos /repeated words, such as 'allow allow', but I felt the structure and the paper was good, the pacing worked well for me, the explanations were clear. I did have a slight hiccup over the use of both PPO and Gumbel, which I eventually realized is because the sender-receiver network as a unit has to operate in an environment which itself gives a stochastic reward, so a small diagram illustrating the relationship between the sender agent, the receiver agent, and the environment/reward might faciltitate this point slightly.

I felt a little odd that the paper both introduces PPO as a way to handle stochastic reward, I mean, relative to other emergent communication papers, and then draws conclusions based only on using PPO. But PPO is a common approach to stochastically rewarding environments, and the conclusions, experimental, and theoretical work, seemed solid to me.

I felt the analogy between ECRP and PPO was slightly stretched to me, but I could more or less see the connection betweeen the two. I think the point I felt least confortable with was comparing positively rewarded episodes in the PPO buffer with the individual 1/beta samples in ECRP. I am unfamiliar with PPO, and would have appreciated some clarification why we are only considering positively-rewarded episodes in this comparison. Does PPO only store positively rewarded episodes in the buffer? Or are considering that only the positively-rewarded episodes will modify the weights, in a particular direction? So, some clarification on your exact thinking on this point, and exactly how PPO works, on this point, would be appreciated please.

## Good points

- use of PPO in emergent communication is new
- empirical exploration of effect of reward shaping on the entropy of the emergent language is new
- theoretical analysis of why the reward shaping affects the entropy of the emergent language is new

## Bad points

- a few typos/repeated words (e.g. 'allow allow')
- could do with a diagram showing the relationship between the sender model, the receiver model, and the task/reward (and in particular how Gumbel and PPO fit into this)

## Questions for author

None. The work seemed rather complete to me.

## Notes

Notes I made whilst reading the paper:

### Abstract

by the end of the abstract I'm excited to see what the paper is going to show.

### Introduction

seems rather specialized (experience buffer size), but if it does provide both empirical evidence and theory, then, ok :)

### 2.1 taxonomy

concept and analysis of 'trivially optimal language' is interesting

table 1: nit: can we make it so we have both columns to be 'yes', rather than one aiming for 'no' and one for 'yes'.
- maybe a way of phrasing trivial in the other way around. ('non-trivial'? :) )

**Summary Of The Paper:**

The paper examines the effect of reward shaping in the context of emergent communication, and specifically with respect to the Shannon entropy of the emerged language. The paper showed that shaping the rewards can lead to higher entropy than training with just the base reward. We care about the entropy because this relates to how expressive the language is. There are existing works, such as Kharatinov et al 2020, which examine the relationship between task/environment and the resulting language entropy.

The paper uses PPO as its main outer-level optimizer, which it uses because the task is a multi-step task which might be challenging to learn with other algorithms such as REINFORCE.

The paper provides theoretical analysis of the effect of reward shaping on entropy by considering an analogy between PPO learning and a process they call the 'Extended Chinese Restaurant Process' (ECRP), where at each step, the customer can be divided into beta parts, and each of the beta bits of customers (which are now each of size 1/beta) takes a seat. This is presented as being analogous to how PPO first store multiple episodes over a buffer, and then updates the weights of the network, using the results of the whole buffer; then rinses and repeats.

Experimental results are presented which show that making the task harder/longer (by increasing the world radius) results in a fall of the entropy of the resultant emergent language, but not in the presence of reward shaping.

The paper asserts that whilst multi-step tasks, and PPO, are not much used in emergent communication literature currently, but it seems plausible that this will become the case in the future, and this paper is a small exploratory step in that direction.

**Summary Of The Review:**

Like the paper
- use of PPO in emergent communication is new
- empirical exploration of effect of reward shaping on the entropy of the emergent language is new
- theoretical analysis of why the reward shaping affects the entropy of the emergent language is new

---

> ### Author Response · Authors · 2021-11-17
> **Clarifying question**
>
> Thank you for your review.
>
> > I felt a little odd that the paper both introduces PPO as a way to handle stochastic reward, I mean, relative to other emergent communication papers, and then draws conclusions based only on using PPO.
>
> It is not quite clear what is meant by "PPO being introduced to handle a stochastic reward".
> It is true that we do not need use PPO for the non-differentiability of the discrete bottleneck since a Gumbel Softmax layer is differentiable.
> The primary reason for using PPO was to handle the multi-step nature of the RL environment; if this is all that is meant by "stochastic reward" then I believe we are on the same page.
> PPO was used not so as to be a novelty, but merely as strong baseline RL algorithm.

---

> > ### Comment · Reviewer_uFB4 · 2021-11-23
> > **Seem to have missed many points I mentioned?**
> >
> > Observations:
> > 1. given I gave the most positive review out of all reviewers, I feel a little odd that you did not acknowledge that, in any way; and indeed provided the shortest response to my review, compared to the more negative reviews.
> > 2. there were a number of concerns I raised, and you only responded to one of them
> > 3. I don't feel that you addressed my issue. I feel you simply re-stated your original claim, in the paper, that PPO is a strong baseline RL algorithm. However:
> >     - the emergent communication community does not typically use PPO; it mostly uses REINFORCE, or else cross-entropy + Gumbel, in my opinion
> >     - therefore, any findings you show in your paper will not directly apply, or apply at all in fact, to the vast majority (or all?) other emergent communication papers
> >     - this makes using PPO feel a little like a 'straw man' to me. "Look, if we do X, that no-one else does, this gives undesirable outcome Y"
> >     - this then puts you on shaky territory, I feel, so far as showing impact to the emergent communication community, which shaky territory aligns with other reviewers feedback to this submission, I feel, stating that you only conducted one small experiment, that might not transfer to other experiments, or be meaningful/useful to other researchers

---

> > > ### Author Response · Authors · 2021-11-29
> > > **Splitting the paper in two**
> > >
> > > Thank you for these clarifications; I definitely see the point that you are making regarding how PPO seems to be out of place.
> > > The original intent for the paper was for PPO to just be an implementation detail, although this was somewhat haphazardly elevated to "more than incidental" by introducing the ECRP.
> > > In light of this, the goal of the paper as submitted was primarily addressing bias in emergent language and not trying to make arguments specific to PPO.
> > >
> > > Your comments make me think that contents of the paper should be split in two and be revised/expanded individually:
> > > 1. With respect to biasing emergent language, make the environment as similar to others as possible, first by using REINFORCE instead of PPO.
> > >     Incorporating other reviewer feedback: use a variety of environments with more quantitative analyses.
> > > 2. With respect to PPO, look at the effects of specifically moving from REINFORCE to PPO and other similar algorithms.
> > >     The ECRP would better fit in a paper describing this as it might account for effects unique to PPO and/or a certain class of RL algorithms.

---

> > ### Comment · Reviewer_uFB4 · 2021-11-23
> > **Is the goal of your paper to make baby steps towards using PPO? Or to show that shaped rewards bias emergent language?**
> >
> > Is the goal of your paper to make baby steps towards using PPO? Or to show that shaped rewards bias emergent language?
> >
> > - if the former, then I feel that you might need to reposition the research questions of your paper, since as it currently stands the abstract says nothing about PPO, and simply states that shaped rewards bias emergent language.
> > - if the latter, then I feel that you should consider use training approaches which align with current commonly used approaches in current emergent communication literature, and I do not feel that using PPO is a common approach in emergent communication literature (indeed, I don't recall a paper that uses it, though I'm sure you can point me to at least one or two exceptions I imagine)
> >
> > I suppose, whilst reading the paper, in my head, I was framing the problem as something like "if we start moving towards PPO, what issues can we encounter, that we should be careful with?". This is perhaps more aligned with the former question, but is not at all reflected by your current title and abstract.
> >
> > I'm going to change my rating to 'marginally above acceptance level'. I do like the paper, a lot, but I feel you might consider more clearly what is the actual research question you are trying to answer, and to what extent using PPO aligns with it. (I actually personally like the PPO approach, but then I feel you should make this more obvious in your title, and in your abstract).

---

### Official Review · Reviewer_2p5d · 2021-11-02

**Correctness:** 3
**Technical Novelty And Significance:** 2
**Empirical Novelty And Significance:** 2
**Recommendation:** 3
**Confidence:** 4

**Main Review:**

My main issue with this paper is that the core contributions may not be particularly useful to people working in emergent communication for different environments and tasks. The general claim that "shaped rewards bias emergent language" seems well known in the literature. Many papers have been using shaped rewards in some form or another to try to bias the language towards certain results—usually the target is trying to increase the systematicity and compositionality of the language, e.g. with auxiliary losses ([Chaabouni et al., 2019](https://arxiv.org/abs/1905.12561); [Luna et al., 2020](https://arxiv.org/abs/2004.03868); [Mihai and Hare, 2021](https://arxiv.org/abs/2106.02067), among many others). This work does not pretend that the shaped rewards are not biasing the language—indeed, **the goal is to bias the language**, and the central question is *what shaped rewards lead to the kinds of qualitative language behavior we desire in emergent communication*.

The authors suggest that shaped rewards obscure our ability to study how emergent languages arise from "basic principles." It's not precisely clear to me what these "basic principles" are (every study likely has some sort of contentious design choice somewhere), but there also seems to be some consensus that languages developed with minimal guidance/shaping are successful, but are generally uninteresting and unhelpful (e.g. [Kottur et al., 2017](https://arxiv.org/abs/1706.08502)), both for people interested in computational simulations of linguistic evolution and for those ultimately interested in connecting emergent languages to real ones.

The authors show that, for a simple multi-agent navigation environment, adding shaped rewards (i.e. auxiliary signals) to the environment bias the emergent language. Granted, since this environment is more slightly more complex than some previous work (insofar as it involves communication over multiple timesteps), so there may be some takeaways for researchers looking at emergent communication in multi-step settings. Such shaped rewards look different than the kinds of regularizers and auxiliary losses used in non-multi-step games, and perhaps the PPO experience buffer observation will be useful for people who use such experience buffers in other studies. Despite this, however, my main concern is that the analyses in this paper seem to have limited utility and applicability outside of the specific navigation environment explored.

It would have been good to see a wider variety of environments/tasks explored, including those that have already been explored in the literature. Otherwise it's unclear what other researchers will get from this paper, unless they're using the exact same navigation task used in this paper (assuming that the insight that "shaped rewards bias emergent language" is already known to them, which I believe is fairly well-established). I believe a more useful version of this paper would show how consistent kinds of shaped rewards (e.g. denser rewards) consistently bias the language across a variety of tasks, thereby substantiating more general claims and considerations that may be useful to people studying multi-agent communication regardless of the environment.

# Strengths

- Some insights into how auxiliary rewards and training parameters bias emergent communication, esp. in RL, may be useful considerations for future researchers. It does paint a worrying picture that something as simple as PPO experience buffer size has dramatic effects on the entropies of the learned languages, which calls into question how stable conclusions drawn from studies of RL communication can be. Though it would be nice to see other environments where these hypotheses also hold, especially environments that have previously been used in multi-agent communication (of which there are now several RL environments have that been explored).

# Weaknesses

- Limited evaluation—no other environments explored.
- The investigation of how shaped rewards change the entropy/semantics of the language is interesting, but seems limited to this environment (especially, for example, with world radii). I don't think it's particularly surprising that shaping rewards change the dynamics of the language, particularly with respect to their sensitivity to features that are specific to the environment (e.g. world radii). In general we must watch out for confounding factors when modifying our environment across all studies in emergent communication. It's not clear to me how the confounding factors in this environment are teaching us something new about how to approach emergent communication studies across all environments.
- I feel like the term "basic principles", "first principles" is thrown around a lot and it's unclear to me what precisely this means. Some more time could be spent explaining what the authors mean, e.g. end of section 2.2 p1, explanation of Bullard et al. 2021 in intro p1,. In fact I expect many design choices made in previous studies on emergent communication argue that their particular reward/choice of game/etc are already "first principles".
- The sea star plots are confusing to me, and more time should be spent explaining them. Why are there only 3-6 arms per plot if there are 64 bottleneck units? Does this mean the teacher does not use the other units to communicate, or do the other units not have interpretable effects on the receiver's trajectory? Does the receiver **always** take an action in a deterministic direction upon seeing? Surely there's some spread/variance—in which case that should be evident in the graph somewhere. Or rather, are all 6 sea star plots responses to different units in a single language? But then the caption "Each sea star corresponds to an independent language" wouldn't be correct...
- Writing seems incomplete in some places, with obvious omissions. Figures should have captions. Intro: "there is a risk that the design choice simply mirrors" repeated.
- "We remove any models which do not [achieve a 100% success rate] from consideration." Without any additional details, this seems highly suspect; ideally you'd find a setting where the agents consistently reach 100% success rate (else the reproducibility statement claim that "the training process is stable" is false); at a minimum, you should describe what proportion of agents fail to converge, and whether the languages in this discarded portion differ qualitatively from the included languages in a way that might bias the results examined here.
- I think some of the specifications in Table 1 are incorrect. E.g. Lazaridou 2018, task is considered "trivial" but Lazaridou et al. also explore a real pixel reference game where the task is to communicate over real images. Authors argue this task is trivial since one can learn the underlying features (of shape and color), but extracting these features is very much a non-trivial task. In the same way, then, the Havrylov and Titov paper can also be considered trivial, in that one should just be able to learn the underlying MSCOCO objects that the dataset is sampled from and those objects.
- The organization of this paper seems off. I feel like section 4 (the CRP) should be introduced around section 5.4, since it is completely ignored until then. The relation between section 5.4 and the rest of the paper could also be made more clear—this investigation seems completely tangential to the question of how shaped rewards bias the environment, and the experiment is not mentioned in abstract.
- Figure 4: the connection between "lower entropy at higher world radii" needs to be better connected with the claim that "agent pair often fails to learn a fully successful language at the highest world radii". What does "fully successful language" mean? Don't agents get 100% task accuracy on this task---then why isn't the language "successful"?
- Following the prev. question, in terms of actually exploring the emergent language, the only metric really investigated here is entropy. In any experiment I could likely find some metric for the resulting language that changes wildly if I modify training hyperparam or environment reward. Authors should therefore more clearly justify why entropy is the only relevant part of the story here. Why is it good or bad that shaped rewards may lead to a language with higher entropy than is expected from "basic principles"? There are a lot of other metrics, e.g. related to compositionality ([Brighton and Kirby, 2006](https://pubmed.ncbi.nlm.nih.gov/16539767/)), or positive signaling/listening; ([Lowe et al., 2021](https://arxiv.org/abs/1903.05168)), that we might care about, and would be good to measure here. Why should we be so concerned/worried about changes to our environment affecting the entropy of the resulting languages?

# Questions

- What does it mean to "emerge from basic principles?"
- Are the experiments in Section 5 done in the edgeward environment, centerward environment, or both? Are there qualitative differences in the behavior of both settings, and if not, what is the motivation for exploring both settings?
- If i'm understanding this correctly, It's interesting that the agent messages encode horizontal commands (Figure 3c) even though horizontal information is given in the biased reward. I would expect that agents learning to encode vertical actions would help fill in the deficiencies of the environment reward. Though presumably the vertical actions are harder to learn.
- How often does the sender send messages to the receiver - at every timestep, or just once at the beginning of each episode?
- Why is the PPO experience buffer experiment an "alternative explanation" for the differing entropies observed in the paper? Couldn't experience buffer size and reward shaping affect entropy (or other language metrics) in orthogonal ways?

# Minor

- 5.4 "model presented in the next section"...what next section?

**Summary Of The Paper:**

This paper studies the effect of auxiliary rewards in an environment for a multi-agent communication task. The task is for a sender to navigate a receiver to the center of some environment, in worlds with varying sizes, and must develop a discrete language (consisting of 1 of 64 symbols) to solve this task.

The authors argue that seemingly innocuous ("unbiased") shaped rewards in the environment have significant, measurable effects on the learned languages, both qualitatively and quantitatively. The authors both visualize the effect of shaped rewards on the emergent language, and measure the change in entropy to the languages caused by the denser shaped rewards. For example, adding denser rewards increases the entropy (and presumably information) communicated by the language; denser rewards prevent a decrease in entropy for languages generated for larger (more difficult) tasks, suggesting that information bandwidth remains high in the shaped reward setting.

Finally, the authors suggest an alternative hypothesis for the differential entropies observed by language in this game: the size of the experience replay buffer in their PPO algorithm, which roughly corresponds to the frequency of updating the semantics of the learned language. There is some connection drawn here to a batched CRP with varying batch size, though it's not super clear to me how strong this connection is to PPO learning and why this CRP is the right way of thinking about experience buffer sizes.

The core argument of the paper, then, is to more carefully consider how environmental rewards might change the semantics of the learned language and mask other factors of interest in studies of emergent communication.

**Summary Of The Review:**

See above.

---

> ### Author Response · Authors · 2021-11-17
> **General rebuttal (1/3)**
>
> # Summary
>
> Thank you for the excellent review which shows a thorough understanding of the paper and of the broader field.
> It appears that the largest cirtique points out the limited scope of the empirical contributions which, in turn, limit the broader applicability of the presented study.
> While the experiments are limited to a single environment, the primary thrust of the paper is a conceptual one, namely that modifications to the environment with an "engineered intent" can bias the learned language in obvious and non-obvious ways.
> Thus, emergent language researchers who trying to scientifically observe emergent properties of a language (as opposed to taking a metric-optimization engineering approach) ought to account for the changes which shaped rewards can introduce.
> We agree with the critique that the discussion of what "natural emergence" or "artificially engineered" is very limited, but we hope to clarify this through the rebuttal below and thereby better demonstrate the import of this paper.
>
> Although we highlight single lines below from the main review, each response is intended to address the whole paragraph.
> If this rebuttal is too long-winded, we can condense it; it is just that there are many good points to respond to.
>
> # Main Review
>
> > My main issue with this paper is that the core contributions...
> > indeed, the goal is to bias the language, and the central question is what shaped rewards lead to the kinds of qualitative language behavior we desire in emergent communication.
>
> A better (albeit long-winded) title for the paper would be, "shaped rewards bias emergent language in non-obvious ways that interferes with the scientific investigation of the language."
> We do agree that many other papers bias the emergent language _on purpose_ in order elicit interesting characteristics.
> This paper, in turn, argues that when this done, it perturbs other properties of the emergent language.
> Thus, if these environments with inductive biases serve as basis for further study, these inductive biases ought to be treated as a confounding factor.
> This is the claim of the paper with respect to the works brought up in this paragraph.
>
> Secondly, the paper also seeks to anticipate future emergent language research which investigates more complicated environments which would incorporate shaped rewards simply in solving the task from a plain RL perspective, let alone eliciting interesting properties in an emergent language.
>
>
> > It's not precisely clear to me what these "basic principles" are (every study likely has some sort of contentious design choice somewhere)...
>
> This is a good question and, generally speaking, far beyond the scope of this paper, though we agree with you that there ought to be more clarity nonetheless.
> At least in terms of the reward, we would argue that "basic principles" entails receiving a reward for the ultimate goal the task and nothing else.
> Any auxiliary rewards (e.g., predicting the other agents' goals in Mordatch & Abeel, 2017) would be beyond "basic principles"; this would not thereby invalidate the experiment, but it should then be considered as a confounding factor and ideally be addressed experimentally.
> This distinction is very obvious (purposely so) in this paper's environment.
> But even when it is less clear in other environments, the primary argument of this paper is that even when it is less clear, the confounding factors of these inductive biases need to be taken into account regardless of whether they are intentionally biasing the language towards compositionality et al. or just to improve the stability of the reinforcement learning process.
>
>
>
> > Despite this, however, my main concern is that the analyses in this paper seem to have limited utility and applicability outside of the specific navigation environment explored.
>
> As stated above, the primary claim of the paper is of type "here is an abstract risk in emergent language research which we are identifying and clarifying, and we back this up with a few concrete experiments" rather than, "here is a gamut of potential biases that can crop up in emergent language which others can directly apply to their own _empirical_ evaluations".
> It is an empirical "existence proof", in a sense.
>
>
> > It would have been good to see a wider variety of environments/tasks explored, including those that have already been explored in the literature.
>
> We agree with advantage afforded both by a wider variety and a use of specific prior work.
> On the other hand, in order to provide a succinct analysis of the quantitative and qualitative effects of reward shaping with as few confounding factors as possible while still working within a "deep" learning paradigm, we selected a handful of empirical evaluations with a single simple environment.
> Further expansion of the environments would entail giving a more cursory analysis of each environment.

---

> > ### Comment · Reviewer_2p5d · 2021-11-22
> > **Thanks for response to review; still feel the conceptual contribution is limited**
> >
> > I appreciate the authors' response to my review, especially clarifying the framing of the paper.
> >
> > > shaped rewards bias emergent language in non-obvious ways that interferes with the scientific investigation of the language." We do agree that many other papers bias the emergent language on purpose in order elicit interesting characteristics. This paper, in turn, argues that when this done, it perturbs other properties of the emergent language
> >
> > Since we agree that it is commonly known that environmental changes bias the emergent language, the key contribution to evaluate here is the paper claiming that biasing "non-obvious" parts of the environment (presumably those that deviate sufficiently from "first principles") changes the language too. Granted, some changes to the environment do not affect the language as directly as e.g. applying some interpretability loss on the language tokens themselves, but who defines how "non-obvious" an environmental change is? For example, here are things that have been toyed around with:
> >
> > - more symbolic input representations result in more compositional communication ([Lazaridou et al., 2018](https://arxiv.org/abs/1804.03984))
> > - adding more agents to your environment changes the communication protocol ([Graesser et al., 2019](https://arxiv.org/abs/1901.08706), [Kim and Oh, 2021](https://proceedings.neurips.cc/paper/2021/hash/92dfa194391a59dc65b88b704599dbd6-Abstract.html))
> >
> > These could similarly be argued to be "non-obvious" changes that may or may not have affected the emergent language (it's obvious in hindsight, now).
> >
> > Overall, the key assumption of the paper—that most researchers do not know that "non-obvious" changes to an environment can bias the emergent language—remains suspect to me, since the community has long been applying arguably subtle changes to an environment to shape the emergent language. Granted, the reward shaping explored here is perhaps more "non-obvious" than existing literature, but this distinction is hazy and only explored in a single environment with limited external validity. Thus, in the current state I'm not sure I find this contribution sufficiently large enough to warrant an entire ICLR paper.
> >
> > Overall my recommendation hinges on the opinion that most researchers
> >
> > 1. would find it unsurprising that the changes to the environment made in this paper bias the emergent language, even if the changes are arguably "less obvious" than existing work;
> > 2. would find limited ways to apply the findings in these experiments to other environments, as there are no broader claims or insights drawn about how rewards bias the emergent language that are applicable outside of this environment.
> >
> > > Further expansion of the environments would entail giving a more cursory analysis of each environment.
> >
> > On the contrary, I would argue that a more cursory analysis of more varied and typical environments, along with some trends that authors see across environments, would actually strengthen the paper.
> >
> > The conceptual claim the authors are trying to make---that "non-obvious" shaped rewards bias language---is a very simple one. And it in fact **needs no more than a cursory analysis of each environment**. I'm not what advantages a large array of experiments, as presented in the paper, give us versus just a single experiment (perhaps explained in a single paragraph or two), for helping us understand the core claim that "shaped rewards bias language." The extensive experiments presented in this paper provide a litany of ways in which the emergent language can be biased, but only give us a deep understanding of this single handcrafted environment, which is of limited use to researchers studying other settings.
> >
> > ---
> >
> > I would like to keep my score at 3, but am happy to hear further from the authors and from the other reviewers if they disagree with my recommendation.

---

> > > ### Author Response · Authors · 2021-11-29
> > > **Reply**
> > >
> > > Thank you for the further response; the perspective you provide on the paper's framing will prove very helpful in future revisions.
> > > I am largely in agreement that the philosophical aspects of the paper (i.e., how to define "non-obvious" and "first principles") as well the empirical ones (i.e., breadth of experiments) could both be greatly strengthened.
> > > Although I hoped for the depth and breadth of this paper to be of sufficient merit, I respect the judgment of the reviewers as to whether the findings are significant or not.

---

> ### Author Response · Authors · 2021-11-17
> **General rebuttal (2/3)**
>
> ## Weaknesses
>
> > I feel like the term "basic principles", "first principles" is thrown around a lot and it's unclear to me what precisely this means. Some more time could be spent explaining what the authors mean, e.g. end of section 2.2 p1, explanation of Bullard et al. 2021 in intro p1,.
>
> Building on the response above regarding "basic princples", we will address the particular example pointed out here.
> At the end of Sec 2.2 p1, this use of "first-principles" is actually distinct from what we mean by "emerging from first principles"; by this, we simply mean to say that using a wide variety of produces a number of confounding factors which makes it more difficult to make a general claim, whereas in this paper, we purposely choose a simple environment so as to minimize confounding factors.
>
> In discussing Bullard et al. 2021 in the introduction, we are primarily arguing that the paper's approach to emergent language is more firmly within the tradition of engineering (instead of science) since rather than starting from agents who learn simply from the reward signal associated with achieving the ultimate goal, the experiment focus on evaluating the effectiveness of discovery algorithm built explicitly for that purpose.
> The scientific type of understanding which seeks to understand the basic principles underlying communication (cf. discovering "laws" in other social sciences like supply and demand in economics) is not amenable to studying the "emergence" of language a handcrafted algorithm as presented in the paper -- this not a bad thing generally, just a mismatch in this case.
>
>
> > The sea star plots are confusing to me, and more time should be spent explaining them.
>
> There are only 3-6 arms because the sender converges to only using 3-6 distinct messages for any given run.
> The messages would probably elicit some action from the receiver, but since the sender does not use them, we do not display them.
> At test time, the receiver selects a deterministic action (i.e., (x, y) vector) based solely on the sender's message; there is stochasticity during the training process as part of Stable Baseline 3's implementation of PPO.
>
>
> > Lack of captions.
>
> These can be added in revisions.
>
>
> > "We remove any models which do not [achieve a 100% success rate] from consideration." Without any additional details, this seems highly suspect; ideally you'd find a setting where the agents consistently reach 100% success rate
>
> You are right that this is not addressed extensively enough.
> The only situation where agents do _not_ consistently reach an evaluation success rate of 100% is at the upper third of "world radii" with no shaped rewards which we find intuitive because in very large world, it is difficult to find the goal by random chance, a precondition for learning without shaped rewards.
>
>
> > I think some of the specifications in Table 1 are incorrect. E.g. Lazaridou 2018, task is considered "trivial" but Lazaridou et al. also explore a real pixel reference game where the task is to communicate over real images.
>
> We argue that Lazaridou et al.'s setup is "trivial" insofar as the underlying concepts, the shape and color of the objects, lends itself to a language with one-to-one correspondence with those underlying concepts.
> In contrast, Havrylov and Titov 2017 use MS-COCO at the individual image level.
> So while the dataset does have 80 categories, the batch size of their model is 128, thereby necessitating that distinguishing between the target image and distractors requires more granularity than just the category level.

---

> ### Author Response · Authors · 2021-11-17
> **General rebuttal (3/3)**
>
> ## Weaknesses (contd.)
>
> > Figure 4: the connection between "lower entropy at higher world radii" needs to be better connected with the claim that "agent pair often fails to learn a fully successful language at the highest world radii". What does "fully successful language" mean? Don't agents get 100% task accuracy on this task---then why isn't the language "successful"?
>
> Maybe this ought to be better signalled visually (and textually), but the right side of Figure 4b is missing dots because those runs failed to develop an emergent language (and policy) which achieves 100% accuracy during evaluation (i.e., they are not "successful").
> Before the this failure, the entropy is decreases more rapidly in the "no shaped rewards" environment than in the "shaped rewards" environment -- this is the primary observation we are trying make here.
>
>
> > Following the prev. question, in terms of actually exploring the emergent language, the only metric really investigated here is entropy. In any experiment I could likely find some metric for the resulting language that changes wildly if I modify training hyperparam or environment reward. Authors should therefore more clearly justify why entropy is the only relevant part of the story here. Why is it good or bad that shaped rewards may lead to a language with higher entropy than is expected from "basic principles"?
>
> Due to the simplicity of this environment, the only metrics that would really apply would be entropy, performance (e.g., number of steps to get to the goal), and something taking into account the semantics of actions (e.g., how uniformly distributed the arms of the sea star).
> Of these options, Shannon entropy is a fundamental property in information theory which specifically deals with abstract properties communication; hence entropy is a strong candidate for a metric which is applicable to most any sort of language.
>
> On the other hand, a metric measuring compositionality would not apply to this paper's setup since the messages are always of length 1, precluding compositionality.
> Secondly positive signalling/listening also does not apply because the receiver takes action solely on the sender's message (i.e., it is "blind" and cannot observe the world for itself), so it is forced to listen in a sense.
>
>
> > Are the experiments in Section 5 done in the edgeward environment, centerward environment, or both? Are there qualitative differences in the behavior of both settings, and if not, what is the motivation for exploring both settings?
>
> Edgeward: 5.1.
> Centerward: 5.2, 5.3, 5.4.
> The centerward environment is arguably more natural insofar as there is a single goal region which must be reached from a variety of starting locations.
> As a result, it minimally requires 3 learned actions in order to span the 2-D space; this pressure means that agents tend to learn qualitatively similar solutions in order to reach the goal.
> The edgeward environment is very trivial since consistently taking (full) steps in any given direction leads to a language with a 100% success rate.
> Since the goal can be reach by travelling towards any edge, it is much more common for the language only to represent a limited subset of the directions since there is far less pressure to learn a widely-distributed set of directions than in the centerward.
>
>
> > ... I would expect that agents learning to encode vertical actions would help fill in the deficiencies of the environment reward. Though presumably the vertical actions are harder to learn.
> How often does the sender send messages to the receiver - at every timestep, or just once at the beginning of each episode?
>
> The sender sends a message at every timestep since the receiver takes an action solely based on the message from the receiver.
> The language doesn't encode vertical actions because one it learns the horizontal actions, those are sufficient to solve the edgeward environment with a 100% success rate, so there is no need to explore into the realm of vertical actions.
>
>
> > Why is the PPO experience buffer experiment an "alternative explanation" for the differing entropies observed in the paper? Couldn't experience buffer size and reward shaping affect entropy (or other language metrics) in orthogonal ways?
>
> We only show correlation and not causation, so it is possible that the mechanism explained in the ECRP is not causitive of entropy change observed with PPO.
> That being said, the trend which ECRP predicts does appear in PPO when we only vary experience buffer size, so we do find this to be a strong argument.

---

### Official Review · Reviewer_5smG · 2021-11-02

**Correctness:** 2
**Technical Novelty And Significance:** 1
**Empirical Novelty And Significance:** 1
**Recommendation:** 1
**Confidence:** 4

**Main Review:**

The question of how inductive biases can affect the solutions of emergent communication algorithms seems interesting and potentially impactful. However, I think the current manuscript has too many fundamental issues that prevent it from making progress on this problem.
1. The contributions are unclear. The current study is conducted in a very informal manner---without a detailed problem description or an explanation of the empirical methodology---which makes it difficult to identify the main claims, understand the connection between the intended claims and the empirical conclusions, and ultimately leads me to question the correctness and significance of the whole study.
2. It is unclear whether the presented data contains a positive result that would make a significant contribution to research community. The experiment section presents data that seem mostly the same between baseline and comparator.
3. The manuscript does not contain enough details to reproduce its results. Several plots are presented in figures 3, 4, and 5 without a detailed explanation of how they were generated or what their implications may be. It feels like readers are supposed to trust they are correct without question.

#### Questions
* What are the specific research questions this work intends to address?
* How is the presented algorithm used in the experiments?
* What purpose does the Chinese Restaurant Process serve in this work?
* How exactly are rewards shaped in the experiments?
* How are the experiments supportive of the general claims about reward shaping and prior information that you wish to make?


**Summary Of The Paper:**

This paper aims to investigate the negative implications of shaped rewards in an emergent communication learning setting. The authors attempt to view PPO as a Chinese Restaurant Process, and although this link has little relevance to their empirical study, various differences with and without the use of shaped rewards are presented from experience in a single, two-dimensional navigation domain.


**Summary Of The Review:**

I think after addressing these questions in earnest, and giving the study a major overhaul, there could be some very interesting findings from this line of questioning, of interest to both the RL and Emergent Communication communities. But given the major issues it currently has, I will not reccomend for this paper for acceptance.

---

> ### Author Response · Authors · 2021-11-17
> **Clarifying comments**
>
> # Summary
>
> Thank you for your review.
> In this rebuttal, we would primarily like to clarify aspects of the paper and the review which might be unclear.
>
> # Main Review
>
> > The contributions are unclear.
>
> At the end of the introduction we explicitly provide two bullet points explaining what the contributions of the paper are.
> Is it the case that these points are phrased unclearly?
> Or is it the case that points are understood but their broader import is not expanded upon robustly enough?
> One aspect of the listed contributions which might not be explicit enough in the paper is that the its impetus is to frame indutive biases (in the form of shaped rewards) as potential confounding factors.
> This problem is somewhat unique to emergent language research becuase the ultimate object of study is the langauge itself, as opposed to simply trying to maximize performance as in traditional RL.
>
> > The current study is conducted in a very informal manner---without a detailed problem description or an explanation of the empirical methodology
>
> In Section 3, we provide an explanation of the environment, agent architecture, optimization procedure, and reward structure.
> We also include specific hyperparameters in an appendix and copy of the source code.
> In Section 5 we give a conceptual explanation of the experiments with further details in the appendix.
> In the penultimate paragraph of the Introduction, we provide a problem statement before outlining the main claims.
> If you could point a few places in the paper where the language is too informal, it would make responding easier.
>
>
> > It is unclear whether the presented data contains a positive result that would make a significant contribution to research community. The experiment section presents data that seem mostly the same between baseline and comparator.
>
> What specifically in the experiments section does not seem to show a significant difference?
> Our paper does not follow a experimental of having a pre-established baseline which we are trying to outperform.
> Instead, we are providing evidence for the claim that the use of shaped rewards in emergent language research needs to be more judicious due to the bias it can introduce to the learned language.
>
>
> # Questions
>
> > What are the specific research questions this work intends to address?
>
> The specific research that is: Do shaped rewards significantly alter the language which emerges from an environment? If so, in what ways and why might this be an issue?
> This is an important question insofar as shaped rewards, which are commonplace in the broader RL literature, have specific implications for emergent language research since the goal is to study fundamentally emergent properties.
>
>
> > How is the presented algorithm used in the experiments?
> > What purpose does the Chinese Restaurant Process serve in this work?
>
> The ECRP algorithm is used in two ways.
> First, it used as a simplfied model of how PPO alternates between collecting rollouts and optimizing the policy based on these rollouts.
> This model is simplified so that one can intuitive reason about the effects of the experience (rollout) buffer size which are themselves significant.
> Second, the algorithm is used to generate Figure 5a.
>
> > How exactly are rewards shaped in the experiments?
>
> The "unbiased" shaped reward is simply the decrease in the distance to the goal at each step (e.g., 1 for moving a full step directly towards the goal, 0 for moving perpendicular, and -1 for moving a full step away).
>
>
> > How are the experiments supportive of the general claims about reward shaping and prior information that you wish to make?
>
> Even in a simple environment with shaped rewards that are seemingly unbiased (i.e., they do not favor learning specific semantics over others), we observe significant changes in entropy.
> Language which emerges from more complex environments with more varieties of shaped rewards is likely to be even more sensitive.
> To a certain extent, the empirical claims are meant to be an existence proof for the reframing of inductive biases.

---

> > ### Comment · Reviewer_5smG · 2021-11-29
> > **Reply**
> >
> > > Is it the case that these points are phrased unclearly?
> > Yes. The bulleted points are vague, e.g. "demonstrating basic ways", most of the second bullet. However, the issue goes beyond the bullets; the contributions were unclear after reading the entire paper.
> >
> > It would help if the contributions were more explicit, and the language connected precisely with the theoretical / empirical support.
> >
> > > If you could point a few places in the paper where the language is too informal, it would make responding easier.
> >
> > The empirical methodology is informal. It lacks a quantitative metric to make comparisons and provide rigorous empirical support. In Figure 3, it's not clear if there are meaningful differences between 3a and 3b, and to what extent 3c differs from 3a and 3b. Figures 4 and 5 are similarly ambiguous, because the comparison seems to be qualitatively about the shape of the entropy distribution(?)
> >
> > > What specifically in the experiments section does not seem to show a significant difference?
> >
> > All the presented data in figures 3, 4, and 5. See my response above.
> >
> > > Our paper does not follow a experimental of having a pre-established baseline which we are trying to outperform. Instead, we are providing evidence for the claim that the use of shaped rewards in emergent language research needs to be more judicious due to the bias it can introduce to the learned language.
> >
> > I suggest rethinking the empirical methodology so you're answering specific questions with comparisons. In general the comparison should be made with respect to some reference point or baseline. The comparison doesn't have to involve performance, but it should involve something that can be measured and controlled.

---

> > > ### Comment · Reviewer_uFB4 · 2021-11-29
> > > **this is really useful feedback, to me, for my own papers :)**
> > >
> > > This is really awesome feedback, to me, for my own papers, I feel :)
> > >
> > > today I learned: ask reviewers questions such as 'is it the case that these points are phrased unclearly?' :)

---

> > > ### Author Response · Authors · 2021-11-29
> > > **Thank you for the clarification**
> > >
> > > Thank you for the clarifications; it helps me to contextualize the review a bit better.
> > > I will be sure to keep these points in mind when revising this work and designing future work.

---

### Official Review · Reviewer_oZZ8 · 2021-11-03

**Correctness:** 3
**Technical Novelty And Significance:** 2
**Empirical Novelty And Significance:** 2
**Recommendation:** 3
**Confidence:** 3

**Main Review:**

Strengths

The key contribution of the paper is the findings on the impacts of shaped rewards to the the emergent language:
1. shaped rewards can explicitly bias the semantics of the learned language;
2. shaped rewards change the entropy of the learned language;
3. shaped rewards masks the potential effects of other environmental variables of interest.

Weaknesses

The findings are only confirmed in a specific setting, a simple sender-receiver navigation game with two specific bias rewards, Euclidean distance and horizontal distance.

I find the use of Chinese restaurant processes as an analogy not necessary and confusing. In some places, the analogy breaks down as stated "The restaurant analogy breaks down here as we would have to say that in each iteration, β customers simultaneously
and independently make a decision ..."


**Summary Of The Paper:**

This paper investigates emergent language research, whose goal is to study language as it emerges from the inherent properties of the environment, language, and agents. This is typically in the context of reinforcement learning such that the phenomena arises from the pressure to maximize reward. This paper considers two types of rewards. One directly comes from the task. The other is shaped rewards which makes learning easier. This paper mainly investigates the impact of shaped reward on the emergent language. Using a simple sender-receiver navigation game, the paper shows that shaped rewards can explicitly bias the semantics of the learned language, change the entropy of the learned language, and mask the potential effects of other environmental variables of interest.

**Summary Of The Review:**

The paper studies an interesting topic, the impacts of rewards on the properties of emergent language. However, I find the contribution is very light and limited in a very specific setting.

It considers only a simple sender-receiver navigation game with two specific bias rewards, Euclidean distance and horizontal distance. The results are also very straightforward and specific to the setting:
1. "Since the standard shaped reward takes both dimensions into account, we do not see any bias in the direction of the learned actions. With the trivially biased reward, though, we see the the learned languages exclusively favor actions near to the horizontal axis. In this setting, nothing explicitly prevents the agents from learning vertical actions, but the fact that horizontal dimensions receive the shaped reward makes those actions easier to learn."

2. "we plot the language entropies against different world radii. In both settings, we observe that entropy decreases as the world radius increases, but the setting with no shaped rewards shows a much more rapid decrease in entropy. We offer one possible explanation for this effect in Section 5.4. When the only reward signal agents have access to is the base reward, they can only accomplish the task by finding the goal randomly at first; as the size of the environment increases, the chance of finding the goal with random movements decreases and the agent pair often fails to learn a fully successful language at the highest world radii."

The current results can be summarized in a workshop paper.

---

> ### Author Response · Authors · 2021-11-17
> **Clarifying statement and question**
>
> # Summary
>
> Thank you for your review.
>
>
> > The findings are only confirmed in a specific setting, a simple sender-receiver navigation game with two specific bias rewards, Euclidean distance and horizontal distance.
>
> The primary intended contribution of the paper is framing inductive biases (namely in the form shaped rewards) as a unique hazard to emergent language research since it can change the learned language, the primary object of study, in non-obvious ways.
> The empirical evaluations, then, are primarily mean to support this claim using a minimal environment as free as possible from confounding factors.
> Expanding the breadth of empirical evaluations would necessarily reduce the depth of our analysis and treatment of the more abstract issues addressed.
>
>
> > I find the use of Chinese restaurant processes as an analogy not necessary and confusing. In some places, the analogy breaks down as stated "The restaurant analogy breaks down here as we would have to say that in each iteration, β customers simultaneously and independently make a decision ..."
>
> Is it the case that the ECRP is confusing as a way to reason and infer properties about PPO and the experience buffers size hyperparameter?
> Or is it the case the analogy of customers at a Chinese restaurant is a confusing way explain the algorithm presented as the ECRP?
> If it is the latter, the Chinese restaurant analogy is widespread and not original to this paper -- if there is a better analogy to explain the algorithm, we would be happy to use it.
> Further clarification would be required to respond to this point of the review.

---

> > ### Comment · Reviewer_oZZ8 · 2021-11-23
> > **still thinks the contributions are not enough**
> >
> > I do not think the "framing inductive biases (namely in the form shaped rewards) as a unique hazard to emergent language research..." changes my view on the contributions. I think Reviewer 2p5d explained this very well. I do not want to repeat here.
> >
> > About what is confusing, it is the latter. I am not sure there is a need to use analogy. Can not you just explain directly how ECRP works?

---

> > > ### Comment · Reviewer_2p5d · 2021-11-23
> > > **Friendly reminder**
> > >
> > > Just a friendly reminder not to assume the gender of an author/reviewer with pronouns! :)
> > >
> > > - reviewer 2p5d

---

### Official Review · Reviewer_jcC3 · 2021-11-03

**Correctness:** 3
**Technical Novelty And Significance:** 2
**Empirical Novelty And Significance:** 2
**Recommendation:** 3
**Confidence:** 3

**Main Review:**

Strengths:
The paper is well written and motivated, the experiments in the paper are performed with rigor. All the concepts are explained in the paper making the work easily understandable.

Weakness:
The argument for using CRP as a way to explain the RL behavior is not convincing for me, since in the former case the action is picking a table, and every action of picking a table that has already been picked leads to a "reward" but in the case of RL, the action doesn't directly correspond to rewards. And the action and state space for both are different.

The analysis and the results obtained with the agent’s entropy and behavior differences can be said for RL problems in general and not just to emergent language learning. I am not familiar if such analysis has been performed on RL in general. Thus similar analysis on the effects could be carried out in RL in general. The related work section also doesn't include literature on analyzing reward shaping [1] .

The paper has some minor grammatical mistakes that can be fixed with revisions.

The description of the model is confusing in the appendix, the author should use the standard method deep learning community describes a network.

[1] M. Grzes and D. Kudenko, "Theoretical and Empirical Analysis of Reward Shaping in Reinforcement Learning," 2009 International Conference on Machine Learning and Applications, 2009, pp. 337-344, doi: 10.1109/ICMLA.2009.33.

**Summary Of The Paper:**

The paper looks into finding effects of reward shaping on emergent language learning with RL. The work shows the difference via analyzing entropy and the behavior of the learned agent. It draws similarities between the Chinese restaurant process and RL. And shows that some of the differences between the behavior of the agent learned with and without reward shaping can be explained by experience buffer size used in the RL learning algorithm. The work uses a new simple navigation task for the study.

**Summary Of The Review:**

- The work has not taken into consideration the existing literature on analyzing reward shaping on RL.
- The validity of CRP as a way to explain the behaviors is well-motivated and supported in the paper.
- Since the experiments are performed in a simple environment, experiments on a complex task should make that paper stronger.

---

> ### Author Response · Authors · 2021-11-17
> **Clarifying comments (2/2)**
>
> # Summary of the Review
>
> > The work has not taken into consideration the existing literature on analyzing reward shaping on RL.
>
> As stated earlier in the review, traditional RL is more concerned with impacts of shaped rewards on learnability and performance whereas this paper is concerned with the nature of representations (i.e., the emergent language) learned by the agents.
> As a result, most of the existing literature tackles orthogonal issues -- if there is some work which we was cited that direct bearing on the contributions of this work, we will be happy to include it.
>
>
> > The validity of CRP as a way to explain the behaviors is well-motivated and supported in the paper.
>
> Is there a "not" missing in this summary bullet?
> This, rather, seems to be what was discussed in the "Weaknesses" section.
>
>
> > Since the experiments are performed in a simple environment, experiments on a complex task should make that paper stronger.
>
> A greater variety of experiments would improve the empirical contributions of this work.
> That being said, the primary intended contribution of this work is to frame shaped rewards, common place in emergent language research and broader RL, as potential confounding factors for the field of emergent language research; the empirical evaluations are merely a way to demonstrate the claim in the small scale with as few confounding factors as possible.
> To introduce a greater number of environments would necessarily reduce the depth of analysis we could perform on the environments and broader issues.

---

> ### Author Response · Authors · 2021-11-17
> **Clarifying comments (1/2)**
>
> # Summary
>
> Thank you for your review.
> The main theme of this rebuttal is drawing a distinction between this paper and previous (and potential) work in more traditional RL.
> The review seems to state that the paper needs a more explicit connection to previous research in tradition RL, although we argue that emergent language research, while using the techniques of RL, tackles a largely different problem: traditional RL is interested in maximizing performance, robustness, and learnability while emergent language research seeks to study the properties of learned communication protocols.
>
>
> # Main Review
>
> > The argument for using CRP as a way to explain the RL behavior is not convincing for me, since in the former case the action is picking a table, and every action of picking a table that has already been picked leads to a "reward" but in the case of RL, the action doesn't directly correspond to rewards. And the action and state space for both are different.
>
> The ECRP is only intended to explain a restricted subset of the behavior we observe: namely the effect we observe when we directly vary the size of the experience buffer and when there is an effective varying of the experience buffer size through the varying of the world radius.
> It is indeed true that the ECRP makes simplifying assumptions regarding the environment, i.e., not modeling its dynamics or reward structure, but these simplifying assumptions insofar as we are introducing a model that can be intuitively reasoned with.
> Adding in the dynamics and reward structure of the environment would make the ECRP would essentially make the algorithm the exact same RL problem that we explore in our full environment.
> Our primary argument for the usefulness of the ECRP (beyond an _a priori_ similarity to the environment) is through the fact that it predicts very closely the trend that we see in varying the experience buffer size in PPO.
>
>
> > The analysis and the results obtained with the agent’s entropy and behavior differences can be said for RL problems in general and not just to emergent language learning.
>
> This point somewhat misses the mark as both the motivations and analyses of this paper diverge from work in more traditional RL.
> First, the analyses concern the discrete communication protocol developed through the learning process as opposed to maximizing the performance or robustness of the learned end-to-end policy, as would be more common traditional RL.
> Second, this is demonstrated explicitly by the use of entropy to analyze an _intermediate_ discrete representation between the sender and the receiver; if this were a typical RL problem with policy with a standard feed-forward network, analyzing the Shannon entropy of intermediate states would not be a cogent analysis.
>
>
> > M. Grzes and D. Kudenko (2009)
>
> Thank you for this suggestion.
> We can include any relevant information in the paper a subsequent revision, although this paper's focus is largely orthogonal to that of our paper.
> Gzes & Kudenko (2009) are primarily characterizing the effects of shaped rewards on the learning dynamics and performance of RL algorithms while this paper is focusing on the potential pitfalls specific to emergent language research which shaped rewards introduce.
> The difference here is one of the primary goal of emergent language is determining the general characteristics of multi-agent communication, i.e., the representations learned more than the actual performance of the agents.
>
>
> > The description of the model is confusing in the appendix, the author should use the standard method deep learning community describes a network.
>
> We are not familiar with the standard way of describing a network.
> Would it be possible to share a link to a paper or other resource which demonstrates this?

---

### Author Response · Authors · 2021-11-17
**Response to common theme in reviews**

First of all, thank you everyone (AC et al. included) for your reviews and work.

Our interpretation of the reviews is that they largely see the intended contribution as _simply_ demonstrating the ways in which shaped rewards can bias emergent language.
In light of this, since the paper only presents a single environment for empirical evaluation, the contributions of this paper lack the requisite significance for acceptance.

While the empirical evaluations are intended to be illustrative, the empirical contribution is intended to be secondary to the methodological contribution to the paper.
This methodological contribution is pointing out that inductive biases (specifically in the form of shaped rewards) present an unique issue for emergent language research since they bias the language (i.e., the object of study) in non-obvious ways.
Hence while some papers purposely introduce biases to learn a language with certain characteristics (e.g., compositionality), it is imperative to also investigate the confounding factors which these inductive biases have.
Along these lines, the empirical support could be thought of being like an "existence proof" for an exploratory claim rather than a broad-coverage array of examples which, though useful, would lie beyond the scope of a paper exploring the topic for the first time.

We understand that reviewed draft of the paper might not provide this framing clearly enough.
But if the paper more strongly frames the argument in terms of this clarified contribution, would this make any difference in the recommendation of the reviewers?

---

> ### Comment · Reviewer_uFB4 · 2021-11-23
> **Personally, I would prefer doubling down on PPO**
>
> > if the paper more strongly frames the argument in terms of this clarified contribution, would this make any difference in the recommendation of the reviewers?
>
> Personally, I would prefer doubling down on PPO, but making this explicit in the title and abstract.
>
> I haven't used PPO much (at all...), and I'm quite interested to learn about it. Why don't other papers use PPO? What things do we need to be aware of if we use it?

---

> ### Comment · Reviewer_5smG · 2021-11-29
> **Framing would not be sufficient.**
>
> > if the paper more strongly frames the argument in terms of this clarified contribution, would this make any difference in the recommendation of the reviewers?
>
> Not for me. The framing was only a single issue among several. Besides making it clear what questions were being researched, the study should also go on to provide support in favor of a hypothesis, connected with its contribution. Furthermore, that contribution should be somewhat novel and potentially impactful to the ICLR community.

---

### Decision · Program_Chairs · 2022-01-20

**Decision:**

Reject

**Comment:**

All reviewers eventually agreed on rejection. The highest scoring reviewer agreed their interpretation of the framing of the paper caused their initial high-score, where as the other reviewers took a totally different view on the papers contribution. The authors agreed that the text of the paper was not clear in this regard. And the high scoring reviewer downgraded their score and suggested a different pitch.

Much of the reviews focused on how the paper includes a single handcrafted environment for empirical evaluation, and missing related work on reward shaping. In the AC's view (and several of the reviewers said this too) the simple observation "non-obvious shaped rewards bias language" indeed begs of a broader study across a variety of environments.

Whether more experiments are needed or if this work can be reshaped such that one existence proof experiment is enough does not need to be resolved here; the paper in its current form needs significant changes.